# Chikungunya and Mayaro Viruses Induce Chronic Skeletal Muscle Atrophy Triggered by Pro-Inflammatory and Oxidative Response

**DOI:** 10.3390/ijms25168909

**Published:** 2024-08-16

**Authors:** Mariana Oliveira Lopes da Silva, Camila Menezes Figueiredo, Rômulo Leão Silva Neris, Iris Paula Guimarães-Andrade, Daniel Gavino-Leopoldino, Leonardo Linhares Miler-da-Silva, Helber da Maia Valença, Leandro Ladislau, Caroline Victorino Felix de Lima, Fernanda Meireles Coccarelli, Claudia Farias Benjamim, Iranaia Assunção-Miranda

**Affiliations:** 1Department of Virology, Instituto de Microbiologia Paulo de Góes, Universidade Federal do Rio de Janeiro (UFRJ), Rio de Janeiro 21941-902, Brazil; mariianasilva.m@gmail.com (M.O.L.d.S.); menezes.fcamila@gmail.com (C.M.F.); romulo.lsneris@gmail.com (R.L.S.N.); irispaulaga@gmail.com (I.P.G.-A.); danielgavinoleopoldino@gmail.com (D.G.-L.);; 2Instituto de Ciências Biomédicas, Universidade Federal do Rio de Janeiro (UFRJ), Rio de Janeiro 21941-902, Brazil; helfarma@yahoo.com.br (H.d.M.V.);; 3National Center for Structural Biology and Bioimaging (CENABio), Universidade Federal do Rio de Janeiro (UFRJ), Rio de Janeiro 21941-902, Brazil; carolvfl@hotmail.com (C.V.F.d.L.); fernanda.meireles@idor.org (F.M.C.); 4Instituto D’Or de Pesquisa e Ensino, Rio de Janeiro 22281-100, Brazil; 5Instituto de Biofísica Carlos Chagas Filho, Universidade Federal do Rio de Janeiro (UFRJ), Rio de Janeiro 21941-902, Brazil; cfbenjamim@biof.ufrj.br

**Keywords:** arthritogenic alphavirus, skeletal muscle, chronic atrophy, inflammation, oxidative stress

## Abstract

Chikungunya (CHIKV) and Mayaro (MAYV) viruses are arthritogenic alphaviruses that promote an incapacitating and long-lasting inflammatory muscle–articular disease. Despite studies pointing out the importance of skeletal muscle (SkM) in viral pathogenesis, the long-term consequences on its physiology and the mechanism of persistence of symptoms are still poorly understood. Combining molecular, morphological, nuclear magnetic resonance imaging, and histological analysis, we conduct a temporal investigation of CHIKV and MAYV replication in a wild-type mice model, focusing on the impact on SkM composition, structure, and repair in the acute and late phases of infection. We found that viral replication and induced inflammation promote a rapid loss of muscle mass and reduction in fiber cross-sectional area by upregulation of muscle-specific E3 ubiquitin ligases MuRF1 and Atrogin-1 expression, both key regulators of SkM fibers atrophy. Despite a reduction in inflammation and clearance of infectious viral particles, SkM atrophy persists until 30 days post-infection. The genomic CHIKV and MAYV RNAs were still detected in SkM in the late phase, along with the upregulation of chemokines and anti-inflammatory cytokine expression. In agreement with the involvement of inflammatory mediators on induced atrophy, the neutralization of TNF and a reduction in oxidative stress using monomethyl fumarate, an agonist of Nrf2, decreases atrogen expression and atrophic fibers while increasing weight gain in treated mice. These data indicate that arthritogenic alphavirus infection could chronically impact body SkM composition and also harm repair machinery, contributing to a better understanding of mechanisms of arthritogenic alphavirus pathogenesis and with a description of potentially new targets of therapeutic intervention.

## 1. Introduction

Chikungunya (CHIKV) and Mayaro (MAYV) viruses are arthritogenic alphaviruses from the Togaviridae family [1]. Both infections are associated with a febrile disease, similar to other arboviruses, distinguished by the induction of intense and disabling muscle and joint pain [2,3,4,5,6]. The prevalence of muscular symptoms in CHIKV and MAYV-infected subjects occurs at high frequency, reaching about 97% and 77% in some reports, respectively [4,7,8]. In addition, patients could develop a chronic phase of the disease, where muscular and articular pain persist for months or even years after the onset of symptoms [9,10,11,12]. It is well established that long-lasting pain correlates with the maintenance of local inflammatory lesions and increased levels of circulating inflammatory mediators despite the resolution of viremia [10,12,13]. However, the mechanism of persistent immune activation and the consequences of chronic inflammation on skeletal muscle physiology are still not elucidated.

Skeletal muscle (SkM) precursor cells and also mature fibers are important targets of arthritogenic alphavirus replication [13,14]. Muscular infection recruits macrophages and T lymphocytes, which occurs in association with bone destruction [15,16,17,18]. Interestingly, the pathogenicity of different epidemic CHIKV strains was positively correlated with the amplitude of replication in the SkM tissue [8,19,20]. Moreover, analysis of muscle biopsies from CHIKV patients with chronic myalgia and also from experimental models of infection suggests that muscle tissue is a site of persistence of viral antigens, including genomic RNA, which may contribute to chronic immune activation [13,15,17,21,22]. Consistent with these findings, mice infected with an engineered CHIKV strain exhibited restricted replication in SkM despite comparable replication in other target tissues, resulting in decreased disease development, including a reduction in footpad swelling and intraosseous inflammation [23]. This experimental evidence shows the crucial role of muscle replication and induced inflammation in the progression of arthritogenic alphavirus-induced disease. 

Chronic inflammation has been associated with the promotion of long-term consequences for SkM structure and physiology [24,25]. Cytokines secreted by muscular and immune cells are key regulators of SkM myogenesis, metabolism, and reparative programs control after lesions; however, they also drive myopathies and functional loss in acute and chronic disease [25]. Previously, we and other research groups characterized the involvement of the innate and adaptive immune response to the restriction of MAYV and CHIKV replication [26,27,28]. Type I interferon signaling and cellular adaptive immune response were essential to restrict replication and promote tissue clearance of CHIKV and MAYV [26,27,28,29,30]. However, they have also been associated with dysregulated muscle proteostasis and muscle mass restoration in other clinical conditions [24]. This highlights the complex virus-muscular cell interactions associated with replication and long-term symptoms that have not been addressed. 

Here, we performed a temporal investigation of MAYV and CHIKV replication in a susceptible wide-type mice model, focusing on the early and late impacts on SkM fibers’ structure, composition, and repair after injury. We found a positive correlation between MAYV and CHIKV replication, the destruction of muscle fiber, fast loss of muscle mass, and inflammatory response that results in chronic muscle atrophy. Furthermore, we showed that muscle mass waste and atrophy could be prevented by blocking TNF and inducing an antioxidant response. Together, our data provide the first evidence that MAYV and CHIKV-induced inflammation promotes muscle protein degradation and also harms SkM repair machinery, resulting in long-term deficits in body SkM composition and maintenance of lesions that could compromise its physiology.

## 2. Results

### 2.1. MAYV and CHIKV Replication Induce Skeletal Muscle Fiber Atrophy

CHIKV and MAYV long-term immune activation and consequences in muscle composition and physiology have been underexplored. Thus, we first conducted a temporal investigation of CHIKV and MAYV replication and clearance at SkM after subcutaneous infection at the left footpad in a young wild-type mice model (Figure 1). After inoculation, both viruses replicate and increase by 4 logs of infectious viral loads within just 1 day post-infection (dpi) at hind limb gastrocnemius, being sustained at similar levels until 4 dpi in CHIKV and 6 dpi in MAYV infection (Figure 1A). MAYV and CHIKV were also detected at high infectious loads in the left quadriceps and paw at 4 dpi but were also detected in other organs, such as the liver, spleen, and brain (Figure 1B). Despite similar levels in the majority of tissues at this time post-infection, the CHIKV load in the right quadriceps was significantly lower (Figure 1B,C) and was cleared from SkM and other non-articular tissues at 8 dpi (Figure 1C). MAYV was still detected in SkM and also for both viruses at the right and left paw but with decreasing loads (Figure 1C). Despite similar replication and viral persistence in mice paws, CHIKV-induced edema was higher and longer sustained than observed in MAYV infection (Figure 1D), indicating that it does not correlate with the amplitude of viral load. A reduction in weight gain was one of the earliest and most striking clinical signs induced by CHIKV and MAYV infection, and it seems to correlate positively with the amplitude of the peak of viral replication at 4 dpi (Figure 1E). Dissecting and weighing the gastrocnemius muscle at this time-point, we observed significant reductions in size and weight, evidencing that MAYV and CHIKV infections induce fast skeletal muscle mass waste (Figure 1F).

Structural SkM analysis at 4 dpi showed an inflammatory process in gastrocnemius muscle, with cellular infiltrate close to the focus of fiber disorganization and necrosis, mainly in CHIKV-infected muscle (Figure 2A). We also examined areas with reduced fiber calibers, which are characteristic of induced muscle atrophy, including leukocyte infiltrate and necrotic areas. Quantification of SkM fiber cross-sectional area (CSA) from images confirmed that MAYV and CHIKV significantly reduced the fiber area after infection (Figure 2B). To characterize the global impact of infection on the muscle mass volume, sections of SkM from the hind limb were analyzed by nuclear magnetic resonance imaging (NMRI) and reconstituted to determine volume (Figure 2C). A comparison between Mock and MAYV-infected representative NMRI section of SkM before and at day 4 after infection (D4) showed that while Mock mice present a robust growth of muscle mass, MAYV induces mass waste and loss of fiber structure (Figure 2C). Reconstitution of muscle volume showed a significant reduction in SkM mass from the hind limb along MAYV infection, mainly at 4 dpi (Figure 2D). We did not observe differences in tibia or fibula volumes (Figure 2E), indicating that bone growth was not affected. 

To determine the possible mechanism involved in acute SkM mass loss, we evaluated the gene expression of two muscle-specific E3 ubiquitin ligases, the muscle atrophy F-box (Atrogin-1) and muscle ring finger protein 1 (MuRF1), both key regulators of muscle atrophy [31,32]. These enzymes are responsible for protein ubiquitination, which drives its breakdown by the ubiquitin–proteasome system (UPS). We found that CHIKV and MAYV infection promoted a significant upregulation of MuRF1 and Atrogin-1 expression in SkM at 4 dpi (Figure 2F). Taken together, these data indicate that MAYV and CHIKV infections promote an imbalance in protein homeostasis by induction of a catabolic process of SkM. 

### 2.2. Skeletal Muscle Atrophy and Genomic RNA Persist in the Late Phase of Infection 

Muscle atrophy persistence could impact muscle development, physiology, and also motor functions, being associated with chronic disease [33]. Late-time analysis of body composition showed that muscle atrophy persists even after infectious viral particle clearance (Figure 3A,B). Together with corporal differences, the dissected SkM weight of gastrocnemius was significantly lower at 30 dpi (Figure 3B). Interestingly, despite viral infectious particles that could not be detected after 15 dpi, CHIKV and MAYV genomic RNAs persist in SkM tissue (Figure 3C). Confirming muscle atrophy persistence, structural analysis reveals intense inflammation and atrophic SkM fibers at 8 and 15 dpi (Figure 3D). In addition, fiber CSA was still reduced 15- and 30-days post MAYV and CHIKV infections. Inflammatory infiltrate was also reduced at 30 dpi, but we still observed fibers with reduced caliber and centralized nuclei, indicating that CHIKV and MAYV infections trigger chronic SkM atrophy. 

### 2.3. Atrogens Activation Occurs in the Early Phase of Infection Associated with Inflammatory Mediators

Inflammatory mediators, such as TNF, IFN-γ, IL-1β, IL-6, and reactive oxygen species (ROS), are one of the most important signals that activate ubiquitin–ligase expression that induces SkM catabolism and atrophy [24,34]. We quantified the expression of these cytokines in the early (4 and 8 dpi) and late-phases (15 dpi) of CHIKV and MAYV infections and observed a significant increase, mainly in TNF and IFN-γ, but also in IL-6 and IL-1β, however only in the early phase (Figure 4A–D). We also investigated the expression of chemokines responsible for the recruitment of inflammatory cells in the infection of arthritogenic alphavirus. Interestingly, while KC/IL-8 and MCP-1 presented higher levels of expression in the acute phase (Figure 4E,F), the RANTES/CCL5 recruitment signal was increased at 8 dpi, and the expression was greater after viral clearance of SkM (Figure 4G). However, the expression of all chemokines analyzed remained significantly increased when compared with the mock at 15 dpi. We also observed a persistent activation of IL-10 and TGF-β expression, mainly in the acute (8 dpi) but also the late-phases (15 dpi) of infection (Figure 4H,I). Quantification of ROS production at 4 dpi indicates that MAYV and CHIKV infection induces an oxidant environment at SkM for atrogen expression (Figure 4J). Finally, an investigation of MuRF1 and Atrogin-1 expression at 8 and 15 dpi indicates that the signal for activation of persistent atrophy occurs in the early phase of infection (8 dpi) but not at 15 dpi, following the acute profile of inflammatory mediators (Figure 4K,L).

### 2.4. Muscle Atrophy Can Be Reduced by Blocking TNF or Inducing Antioxidant Pathways

Despite its long duration, the mechanisms associated with the activation of SkM atrophy seem to be triggered by inflammatory mediators induced in the acute phase of infections. So, we next tested two pharmacological approaches, first neutralizing TNF-α with infliximab (IFX) and also inducing an antioxidant environment by activation of the nuclear factor erythroid 2-related factor (Nrf2) with monomethyl fumarate (MMF), both approved for clinical use, to control the SkM mass waste induced by arthritogenic alphavirus infection. In agreement with the involvement of TNF in the induced atrophy, we observed that daily treatment with IFX after CHIKV infection reduces muscle mass waste since it increases body weight gain and SkM weight, together with a reduction in levels of Atrogin expression in infected mice (Figure 5A–C). Viral load in SkM at 4 dpi was similar in untreated and IFX-treated mice, indicating that it did not affect viral replication (Figure 5D), and left paw swelling induced by CHIKV was not modified (Figure 5E). 

The MMF treatment was initiated 4 h before CHIKV and MAYV inoculation and also daily in the following days of infection until the day before analysis. We observed an improvement in body weight gain since 2 dpi with MMF treatment, a higher SkM weight, and a reduction in the levels of atrogens, MURF1, and Atrogin-1, mainly in CHIKV infection (Figure 6A–D). Finally, structural analysis of muscle at 8 dpi reveals a preserved muscle fiber structure, together with a global reduction in SkM inflammatory infiltrate (Figure 6E). In addition, we observed reduced areas of atrophic fibers, consistent with an increase in fibers CSA, indicating an efficiency control of SkM atrophy (Figure 6E,F). Interestingly, treatment with MMF also promoted an improvement in viral load control of CHIKV at 4 dpi and of MAYV at 8 dpi (Figure 6G).

## 3. Discussion

Arthritogenic alphavirus infections are responsible for a musculoskeletal disease in humans that, despite low lethality, induces persistent debilitating symptoms, even incapacitating to work [2,35,36]. Due to its epidemiological history, global distribution, and current number of cases in tropical regions [37,38,39], CHIKV has been one of the most studied members of this group. However, these viruses share similarities and differences in the mechanism of promoting tissue lesions that need to be individually addressed for better comprehension of their pathogenesis [18,40]. Despite the robust characterization of immune cells and mediators involved in joint disease progression [17,30,41,42,43], the acute and chronic impact of the inflammatory process on muscle cell physiology was poorly investigated. At present, we have conducted a temporal analysis focusing on the SkM inflammatory disease in mice caused by CHIKV and MAYV infection. We consistently demonstrated that viral replication and inflammatory mediators trigger fast and intense muscle mass loss, resulting in SkM acute and chronic atrophy (Figure 7). Our data significantly contribute to a better understanding of the long-term consequences of arthritogenic alphaviruses on SkM physiology, especially of MAYV infection, a neglected virus from the American continent with increasing and recent expansion into urban areas [44,45]. 

Skeletal muscle is largely abundant and distributed in tissues throughout the body, playing a crucial role in locomotion and regulating body metabolism. SkM growth in young mice occurs mainly through a process of hypertrophy that drives individual muscle fiber enlargement and mass gain [46]. We previously demonstrated that immunocompetent 12-day-old mice are susceptible to MAYV infections, and, consistent with that observed in humans, high viral loads and inflammatory lesions are observed in SkM and joint tissue [26]. Thus, using this model, it was possible to investigate the long-term consequences associated with acute and intense replication at SkM. In addition to swelling on infected paws, a reduction in weight gain was one of the earliest clinical signs of CHIKV and MAYV infections, indicating an impact on protein homeostasis. Using different morphological analyses of SkM at 4 dpi, we demonstrated that infection promotes a reduction in muscle mass volume by fiber destruction and also by shrinking, decreasing muscle mass, and a reduction in fiber cross-sectional area, characteristics of atrophy [25,33,47]. Muscle atrophy is an acute or chronic pathological condition induced by infectious and noninfectious processes, such as aging and cancer, that results in an imbalance of protein synthesis and degradation [25]. In agreement, a study by Ozden in 2007, evaluating quadriceps biopsy of CHIKV patients with myositis, demonstrated evidence that the infection can promote vacuolization and muscle fiber atrophy [13]. The high incidence of muscular symptoms in other arboviruses, although muscular atrophy has not been evaluated, may indicate that it could be a common feature of viral infections [48]. In addition to CHIKV, SARS-CoV-2 infection was the only one in which acute muscle weakness was associated with histological findings of atrophy in patient biopsy [49], reinforcing our hypothesis, and it should be a focus of future studies. 

The persistent atrophy induced by CHIKV and MAYV infections demonstrated by our study in mice models could result in weakness, fatigue, reductions in global mobility, and also metabolic disorders [33]. These same symptoms are largely observed in patients who progress to the chronic phase of arthritogenic alphavirus infection; however, the reduced post-infection quality of life has not been directly associated with muscular atrophy [50]. Interestingly, symptoms are usually more intense in elderly patients, who, in addition to having previous rheumatic complications, also present a chronic loss of muscle mass [51,52]. Similar to the elderly population, the increased susceptibility to viral infection and severity of disease in young mice has been attributed to an inefficient antiviral response [53,54]. This evidence demonstrates the translation of our data with what has been observed in the clinic of the arthritogenic alphavirus, highlighting the importance of understanding the molecular mechanisms enrolled in the induction and persistence of SkM atrophy.

The destruction of fibers and precursor cells by viral replication could trigger a local inflammatory response that may contribute to restoring muscle physiology or to progress to persistent myopathies [21,23]. Increased muscle protein degradation can be mediated by several mechanisms, including activation of UPS and autophagy-lysosome machinery, which may contribute to rapid muscle mass loss [24,47]. Some studies in vitro using a cell culture model demonstrated that activation of UPS and autophagy are essential for efficient CHIKV replication [55,56,57]. In our mouse model, we found that CHIKV and MAYV replications in SkM enhanced the expression of muscle-specific ubiquitin–ligase MuRF1 and Atrogin-1, which are involved in the promotion of protein degradation by UPS and its upregulation drives muscle atrophy [31,32]. Curiously, these atrogens were induced only in the early phase of infection, but reduced caliber fibers persisted until adulthood (Figure 7). The same profile of early upregulation was observed for inflammatory mediators, such as TNF, IFN-γ, IL-6, and ROS, which are well-established activators of muscle degradation by the induction of MuRF1 and Atrogin-1 expression [24,25]. Supporting the association of this inflammatory mediator on induced atrophy, the blockade of TNF, and also the reduction of oxidative stress by using an agonist of Nrf2 during early infection decreases atrogen expression and atrophic fibers while increasing weight gain. 

The rate of body weight gain in infected mice increases after viral clearance from SkM at 8 dpi, indicating a close relation of muscle mass lost with viral replication. Despite temporally increasing CSA, the difference between CHIKV/MAYV and mock-infected mice remains stable, indicating that muscle mass composition was not restored over time. The persistence of viral RNA or proteins in SkM, even in the absence of infectious viral particle assembly, has been raised as possibly responsible for chronic immune activation in arthritogenic alphavirus infection [18]. The MAYV and CHIKV RNA levels were reduced from the early phase but were maintained at detectable levels in the SkM at 30 dpi, confirming that SkM could be a reservoir of viral antigen persistence. Antigen persistence could be a way to maintain signals for muscle degradation or reduce muscle repair machinery, but future investigations are necessary to confirm this hypothesis. A previous study conducted by our group demonstrated that MAYV-induced muscle damage seems to be dependent on lymphocyte infiltrates [26]. Increased levels of CCL5 at 15 dpi could result in the persistence of cellular recruitment signals, such as T-lymphocytes, for SkM [58]. However, this increase, together with the increased levels of IL-10 and TGF-β expression, could also be a shift in the activation of the muscle repair cascade [59,60]. 

The absence of increased levels of atrogens in the late phases of infection indicates that other pathways could be involved in the persistence of atrophy and also needs to be elucidated. Myogenesis is the main process responsible for muscle repair and recomposition after injury, controlled by a balanced pro-inflammatory and followed by an anti-inflammatory immune response [60]. It was demonstrated that ZIKV replication in SkM of neonate mice and also in myoblast cells impairs myogenesis [61,62]. Increased levels of TGF-β have been involved in delays of reparative myogenesis [60]. Thus, since myoblast and myotubes are permissive to arthritogenic virus infection [21], it is possible that chronic-induced atrophy could also be triggered by a delay in muscle repair by dysregulation of muscle myogenesis. 

Treatment of CHIKV and MAYV-infected mice with MMF not only reduced atrophy but also reduced viral replication, indicating that an antioxidant environment does not seem to be favorable for viral replication. MMF is a primary metabolite of dimethyl fumarate that was approved for clinical use in multiple sclerosis patients [63]. In addition to other biological functions, MMF stabilizes and activates the Nrf2 transcription factor, inducing the expression of many target genes coding for antioxidant enzymes, such as HO-1, NAD(P)H:quinone oxidoreductase 1, and proteins for glutathione synthesis [64]. Our data showed an efficient reduction in muscle mass loss, with a significant restoration of fiber CSA and reduction of SkM inflammation and lesion at 8 dpi. Since treatment also reduced viral replication, the MMF effect could not only be attributed to the impact of reduction on ROS, but it can be a consequence of both. Thus, our data indicate that the use of antioxidants or the induction of this pathway to promote a less oxidative environment in the acute phase of infection may be a therapeutic strategy for reducing viral load and also the long-term impact of the infection on muscle composition, which should be better explored. 

## 4. Materials and Methods

### 4.1. Virus Propagation

Mayaro virus (ATCC VR 66, strain TR 4675) and Chikungunya virus (BHI3745/H804709, was kindly provided by Dr. Amilcar Tanuri) were propagated at BHK-21 (ATCC-CCL-10) and in C6/36 cells, respectively, using a multiplicity of infection (MOI) of 0.01. After 30 h of infection culture medium from each infection was collected and centrifuged at 2000× *g* for 10 min to remove cellular debris and stored as aliquots at −80 °C. The viral titer of the stock was determined by plaque assay. 

### 4.2. Mice Infection and Treatment 

Wild-type about 12-day-old SV129 mice of 5.8–6.5 g were subcutaneously infected with 10^5^ or 10^6^ pfu of MAYV or CHIKV in the left footpad, using a final volume of 20 µL. The same volume of virus-free cell cultured medium (Mock) was used as the control. Each experimental group was housed individually in polypropylene cages with free access to chow and water. Young mice were housed with the uninfected mother during all the experiments. For TNF blockade, mice were treated with Infliximab (IFX—Remicade; Janssen-Cilag, Zug, Switzerland) immediately after infection and daily via intraperitoneal (i.p.) inoculation of 20 µg or the same volume of vehicle (PBS—137 mM NaCl, 10 mM sodium phosphate, 2.7 mM KCl, pH 7.4). Treatment with Monomethyl fumarate (MMF—Sigma-Aldrich, St. Louis, MO, USA) was carried out 4 h before Mock or virus infection, by i.p. inoculation of 20 mg/kg or vehicle (same % of DMSO) via i.p. and one dose daily in the following days. For IFX and MMF treatments, a daily dose was administered until the day before the analysis point indicated in the figures. Experimental groups were monitored for clinical signals, weight gain, and paw edema (left), which was measured using a digital caliper. Tissue samples were collected on the desired days post-infection and stored at −80 °C until processed or fixed in 4% formaldehyde for future analysis. 

### 4.3. Ethics Statement

All experimental procedures performed were in accordance with protocol and standards established by the National Council for Control of Animal Experimentation (CONCEA, Brazil) and approved by the Institutional Animal Care and Use Committee (CEUA) from Universidade Federal do Rio de Janeiro (protocol n° A04/22-036-18, CEUA-UFRJ, Rio de Janeiro, Brazil).

### 4.4. Virus Quantification

MAYV and CHIKV titers in tissue samples and cell cultures were determined by plaque assay in Vero cells. Samples were homogenized in DMEM (#11995065, Gibco-Thermo Fisher Scientific, Grand Island, NY, USA) using a fixed relation of mass/volume (*w*/*v*) and, after a 10-fold serial dilution, were used to infect confluent Vero cells seeded in 24-well plates. After 1 h of adsorption, the medium was removed, and 2 mL of 1% carboxymethylcellulose (*w*/*v*) (#C4888, Sigma-Aldrich) in DMEM with 2% fetal bovine serum (FBS, #12657029, Gibco-Thermo Fisher Scientific, South American, Brazil) was added, and cells were incubated at 37 °C. After 48 h, cells were fixed using 4% formaldehyde, and plaques were visualized by staining with 1% crystal violet in 20% ethanol. The viral titer was calculated as plaque-forming units per mL (pfu/mL) and converted to pfu/g of tissue.

### 4.5. Muscle Structural Analysis

For hind limb SkM weight, mice tendon to tendon gastrocnemius muscle was carefully dissected and immediately weighed. For histological analysis, gastrocnemius was collected at defined days post-infection and fixed with 4% formaldehyde for 24 h. Tissues were embedded in paraffin after dehydration. Paraffin-embedded tissue sections of 5 μm were prepared and stained with hematoxylin and eosin (H&E). Images were obtained using optical inverted microscopy with an objective of 10× (Olympus IX81, Tokyo, Japan) and analyzed using ImageJ 1.52a software for quantifications of muscle fiber cross-sectional area (CSA). Briefly, the area of individual fibers of SkM cross sections was calculated after determinations of image pixel/µm converting scale using microscope image scale. The average fiber CSA of a field was taken from at least 4 animals per condition.

### 4.6. Nuclear Magnetic Resonance Imaging (NMRI)

Muscle volume from the hind limb, along with viral infection, was accompanied by an NMRI performed in collaboration with the National Center for Structures and Bioimaging (CENABIO-UFRJ). Images of skeletal muscle cross sections of Mock and MAYV infected mice were acquired using a 7.0 Tesla MRI system (Varian 7T scanner, 210 mm horizontal bore, Agilent Technologies, Palo Alto, CA, USA). The areas of the section were combined for muscle volume reconstitution (mm^3^), along with the time for each mouse. The same procedure was used for bone analysis. 

### 4.7. Gene Expression and Viral RNA Quantification

Hind limb gastrocnemius muscles were homogenized in DMEM (#11995065, Gibco) using a fixed relation of 0.2 mg of tissue/μL, and 200 μL of the homogenate was used for RNA extraction with Trizol (#15596018, Invitrogen) according to the manufacturer’s instructions. Purity and integrity of RNA were determined by the 260/280 and 260/230 nm absorbance ratios. One microgram of isolated RNA was submitted to DNAse treatment (Ambiom^TM^ DNase I, Thermo Fischer, Waltham, MA, USA) and then reverse-transcribed using the High-Capacity cDNA Reverse Transcription Kit (Applied Biosystems, Waltham, MA, USA). Quantification of cytokines expression was performed using a real-time PCR analysis using Power SYBR kit (Applied Biosystems; Foster City, CA, USA). Cycle threshold (Ct) values were normalized to a housekeeping gene and analyzed using the ΔΔCt method to generate fold change values (2^−ΔΔCt^). For the detection of genomic RNA of CHIKV and MAYV real-time PCR analysis was conducted using TaqMan Mix kit with specific dyes (#4304437, Applied Biosystems, Foster City, CA, USA). A standard curve using viral stocks was constructed to determine CT of the limit of genomic RNA positive samples. Primer and TaqMan dyes sequences are described in Table 1.

### 4.8. Quantification of Reactive Oxygen Species

Left hind limb gastrocnemius of mock and infected mice were dissected at 4 dpi and gently homogenized in cold DMEM (#11995065, Gibco; 1:10 *w*/*v*) using a glass tissue grinder as previously described [65]. Reactive oxygen species (ROS) was determined using 2 µM of 5-(and-6)-chloromethyl-2′,7′-dichlorodihydrofluorescein diacetate acetyl ester (CM-H2DCFDA—#C6827, Invitrogen) at a 1:1 mixture of homogenate/DMEM. DCF fluorescence was measured at the time of addition and after 30 min of incubation in the dark at 37 °C in a fluorescence microplate reader (VICTOR™ X3—PerkinElmer, Shelton, CT, USA), at 492–495/517–527 nm, as recommended by manufacturers. The absolute values of basal fluorescence of each sample were discounted from that obtained at the end of the DCF incubation and plotted as fold change from the media of Mock values. 

### 4.9. Statistical Analysis

Comparisons between the MAYV and CHIKV groups were performed using multiple *t*-tests, and statistical significance was determined using the Holm–Sidak method. For comparison involving the Mock group, we used one-way or two-way ANOVA followed by Tukey’s multiple comparison tests, as indicated. All tests were performed using Graph Pad Prism version 6.00 for Windows, Graph Pad Software, La Jolla, CA, USA.

## Figures and Tables

**Figure 1 ijms-25-08909-f001:**
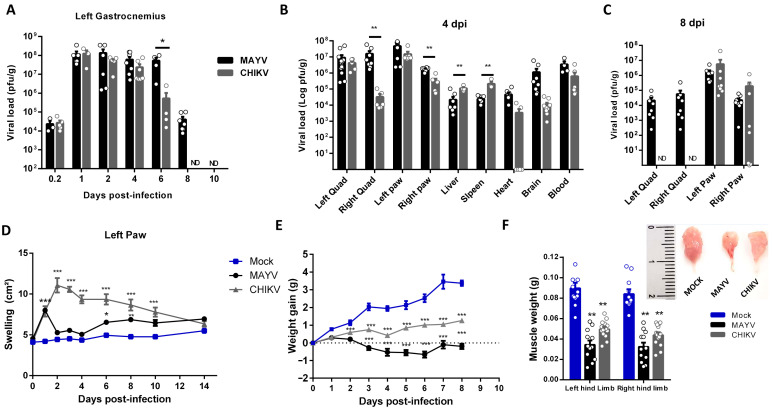
Temporal investigation of CHIKV and MAYV replication and clearance from SkM after subcutaneous infection in a young wild-type mice model. Wild-type (WT) SV129 mice of 12 day-olds were subcutaneously infected with MAYV or CHIKV in the left footpad, and tissues were collected at indicated time points. (**A**) Temporal quantification of viral load by plaque assay in the left gastrocnemius, (**B**) distribution in other tissues with detected infectious particles at 4 dpi and (**C**) at 8 dpi. (**D**) Virus- and Mock-infected mice swelling area of left paws and (**E**) weight gain was accompanied temporally. (**F**) Gastrocnemius muscle was dissected and immediately weighed at 4 dpi. Values were plotted as mean ± standard error of the mean (SEM). The inset shows a representative image of dissected muscles. Statistical analyses were performed to compare (**A**–**C**) viral load of MAYV and CHIKV groups by multiple *t*-tests, and significance was determined using the Holm–Sidak method; (**D**,**E**) swelling and weight gain curve by two-way ANOVA and (**F**) muscle weight by one-way ANOVA followed by Tukey’s multiple comparison test from MAYV and CHIKV groups with mock. * *p* < 0.05, ** *p* < 0.01 and *** *p* < 0.001. Quad = quadriceps muscle. ND: not detected.

**Figure 2 ijms-25-08909-f002:**
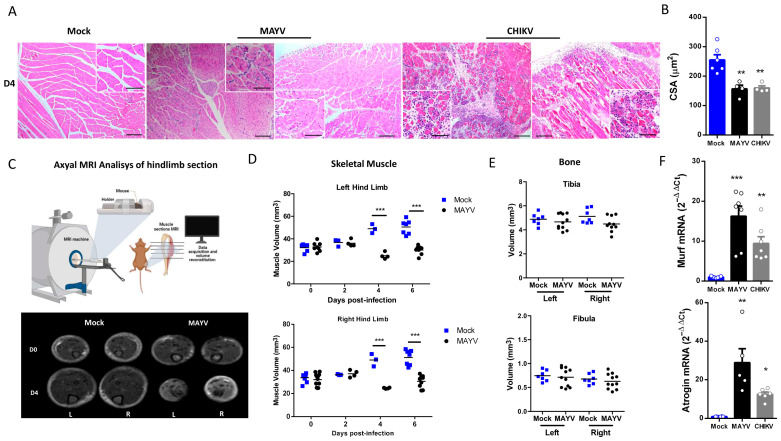
MAYV and CHIKV replication induces inflammatory infiltration and lesions, muscle mass loss, and fiber atrophy in skeletal muscle. (**A**) Left gastrocnemius of Mock or infected groups were collected at 4 dpi and stained with hematoxylin and eosin (H&E). Representative images from 2 infected mice demonstrate the inflammatory cell infiltration, fiber destruction, necrosis, and atrophic fibers in MAYV- and CHIKV-infected animals. High magnification images inset highlight the regions of atrophic fibers. Scale bars of figure = 100 μm and inset 50 μm. (**B**) Quantification of SkM fibers cross-sectional area (CSA) from H&E stained muscle images was performed using ImageJ 1.52a software. Each dot corresponds to the average CSA of fibers present in a field per mouse. (**C**–**E**) The hind limbs of Mock- and MAYV-infected mice were imaged by NMRI. At each time point, images from five sections were acquired to calculate the area and then combined for volume reconstitution. (**F**) Expression levels of MuRF1 and Atrogin-1 in the left gastrocnemius at 4 dpi were determined by real-time PCR analysis. Cycle threshold (Ct) values were normalized to a housekeeping gene and analyzed using the ΔΔCt method to generate fold change values (2^−ΔΔCt^). Values are shown as mean ± standard error of the mean (SEM). Statistical analyses were performed using one-way ANOVA followed by Tukey’s multiple comparison tests (**B**,**F**), and for the comparison of tissue volumes of Mock and MAYV groups, multiple *t*-tests were used, and the significance was determined by the Holm–Sidak method. * *p* < 0.05, ** *p* < 0.01 and *** *p* < 0.001. Image (**C**) was created with BioRender.com.

**Figure 3 ijms-25-08909-f003:**
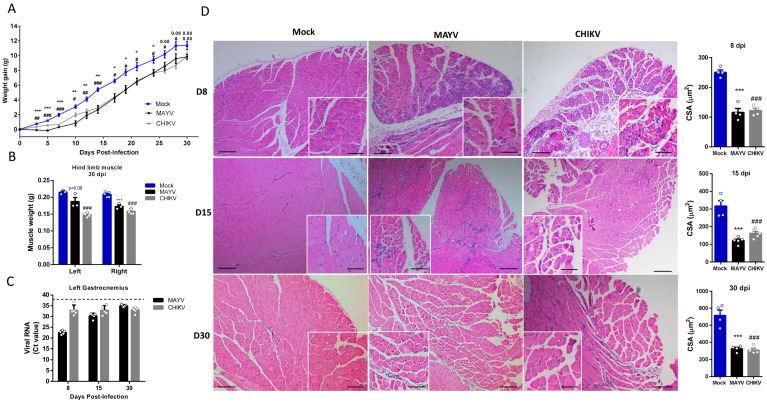
Skeletal muscle atrophy and genomic RNA persist in the late phase of MAYV and CHIKV infection. Wild-type (WT) SV129 mice of 12-day-olds were infected, and SkMs were analyzed at late times post-infection. (**A**) Body weight was recorded until 30 dpi, and (**B**) gastrocnemius muscles were dissected and immediately weighed. (**C**) CHIKV and MAYV RNA were detected by TaqMan real-time PCR analysis. The dotted lines represent the Ct value limit for positive samples. (**D**) Temporal histological analyses of left gastrocnemius stained with H&E and respective quantification of SkM fiber cross-sectional area (CSA) from H&E stained muscle images using ImageJ 1.52a software. Each dot corresponds to the average CSA of fibers present in a field per mouse. Scale bars of figure = 100 μm and inset 50 μm. Values are shown as mean ± standard error of the mean (SEM). Statistical analysis of the weight gain curve comparing MAYV and CHIKV groups with Mock was performed by two-way ANOVA (**A**); for muscle weight (**B**) and CSA (**D**), one-way ANOVA was performed, followed by Tukey’s multiple comparison test. * *p* < 0.05, ** *p* < 0.01 and *** *p* < 0.001 for Mock and MAYV comparison; *^#^ p* < 0.05, ^##^ *p* < 0.01 and ^###^
*p* < 0.001 for Mock and CHIKV comparison.

**Figure 4 ijms-25-08909-f004:**
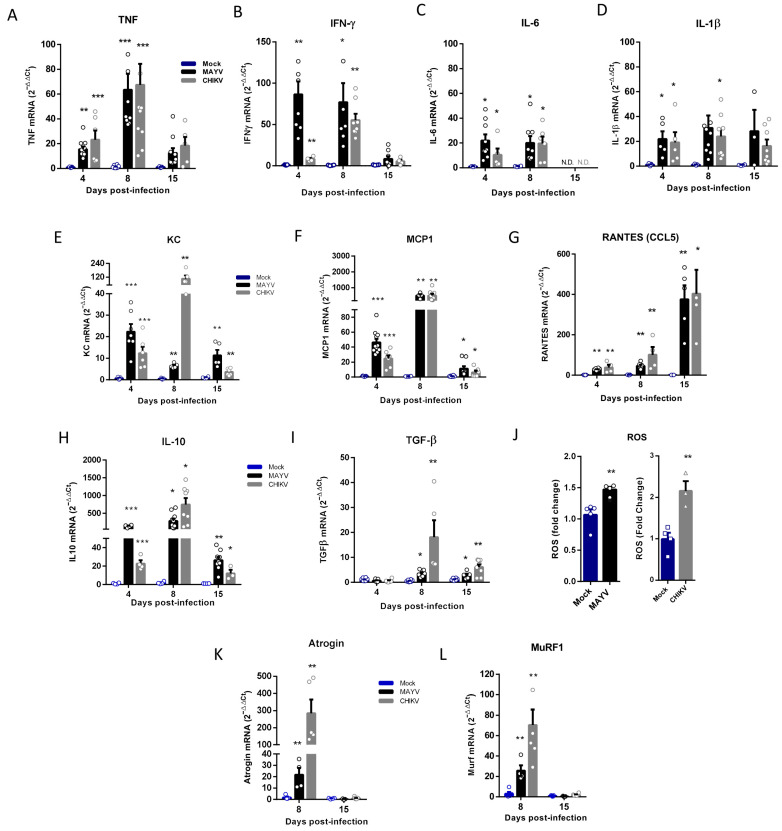
Temporal analysis of inflammatory mediator and atrogen expression in the early and late phases of MAYV and CHIKV infection. Wild-type (WT) SV129 mice of 12 day-olds were subcutaneously infected with MAYV or CHIKV in the left footpad, and left gastrocnemius were collected at indicated time points. (**A**–**I**,**K**,**L**) Quantification of gene expression using real-time PCR analysis. Cycle threshold (Ct) values were normalized to a housekeeping gene and analyzed using the ΔΔCt method to generate fold change values (2^−ΔΔCt^). (**J**) Total reactive oxygen species (ROS) production in the left gastrocnemius was determined by fluorescence analysis using DCFDA. Arbitrary values of fluorescence from each sample were obtained at the end of the DCF incubation and plotted as fold change from the media of mock values. Values were plotted as mean ± standard error of the mean (SEM). Statistical analyses were performed by multiple *t*-tests to compare Mock with CHIKV and MAYV groups, and the significance was determined using the Holm–Sidak method. * *p* < 0.05, ** *p* < 0.01, and *** *p* < 0.001.

**Figure 5 ijms-25-08909-f005:**
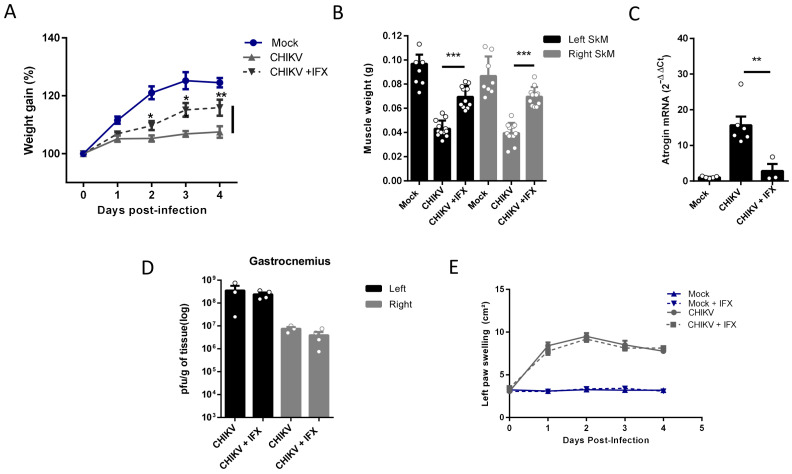
Treatment with Inflixmab reduces muscle mass loss and atrogen expression induced by CHIKV infection. Wild-type (WT) SV129 mice of 12 day-olds were subcutaneously infected with CHIKV and then treated with a daily dose of 20 μg of Infliximab (IFX) or Vehicle (PBS) by i.p. inoculation. (**A**) Weight gain was accompanied daily, and (**B**) gastrocnemius was dissected at 4 dpi for muscle weight measurement, (**C**) Atrogin expression by real-time PCR, and (**D**) viral load by plaque assay. (**E**) Virus- and Mock-infected mice swelling area of left paws was measured daily. Values were plotted as mean ± standard error of the mean (SEM). Statistical analyses of weight gain curve and paw swelling comparing treated and untreated CHIKV infected groups were performed by two-way ANOVA (**A**,**E**); and for muscle weight (**B**), Atrogin-1 expression (**C**), and viral load (**D**), by one-way ANOVA followed by Tukey’s multiple comparison test. * *p* < 0.05, ** *p* < 0.01 and *** *p* < 0.001.

**Figure 6 ijms-25-08909-f006:**
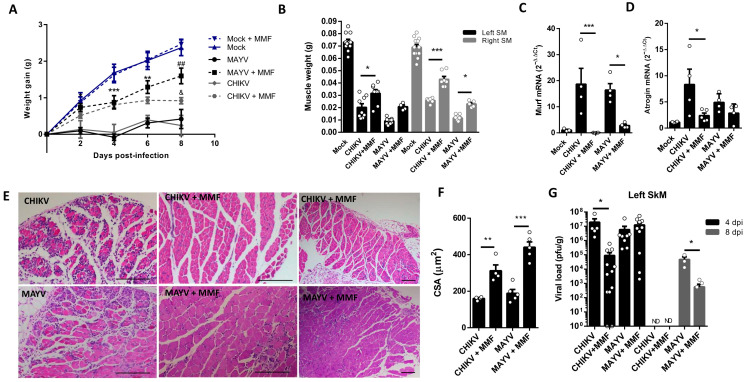
Treatment with MMF reduces muscle atrophy and inflammation induced by MAYV and CHIKV infection. Wild-type (WT) SV129 mice of 12 day-olds were intraperitoneally treated with 20 mg/kg of monomethyl fumarate (MMF) or Vehicle (DMSO) and after 4 h infected with MAYV or CHIKV in the left footpad. (**A**) Weight gain was accompanied daily. Gastrocnemius were dissected at 8 dpi for analysis of (**B**) muscle weight, (**C**) MuRF1, and (**D**) Atrogin-1 expression by real-time PCR. (**E**) Histological analysis of left gastrocnemius stained with H&E (Scale bars of figures = 100 μm) and (**F**) respective quantification of SkM fiber cross-sectional area (CSA) from muscle images using ImageJ 1.52a software. (**G**) Viral load at SkM was determined by plaque assay at 4 and 8 dpi. Values were plotted as mean ± standard error of the mean (SEM). ND: not detected. Statistical analyses of weight gain curve comparing treated and untreated CHIKV and MAYV infected groups were performed by two-way ANOVA (**A**) with * used for indicates significance of comparison of both MAYV and CHIKV groups (4 and 6 dpi); # Only MAYV, and & CHIKV groups (8 dpi); one-way ANOVA was used in the analysis of muscle weight (**B**), Atrogens expression (**C,D**), CSA (**F**), and viral load (**G**), followed by Tukey’s multiple comparison test. */^&^ *p* < 0.05, **/^##^ *p* < 0.01 and *** *p* < 0.001.

**Figure 7 ijms-25-08909-f007:**
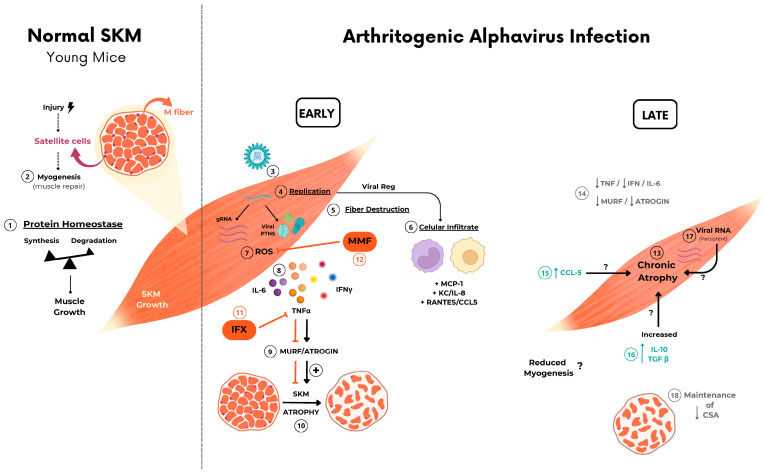
Schematic representation of the possible mechanism involved in arthritogenic alphavirus-induced early and late skeletal muscle atrophy. ① At physiological conditions, young mice SkM fiber growth throughout life is determined by individual muscle fiber enlargement and mass gain by a higher rate of synthesis than protein degradation. ② New fiber generation by myogenesis will be recruited mainly after injury by activation of quiescent satellite cells. ③–⑧ Arthritogenic alphavirus replicates in SkM fibers, resulting in fiber destruction, recruitment of immune cells, and the production of inflammatory mediators in the early phase of infection. ⑨–⑩ Inflammation-induced protein degradation by UPS through activation of MuRF1 and Atrogin-1 that drives acute SkM atrophy. ⑪–⑫ Treatment with IFX or MMF reduces protein degradation and atrophy, corroborating the involvement of these pathways in MAYV- and CHIKV-induced atrophy. ⑬ SkM atrophy persists in the late phase of infection. ⑭ Despite the decrease in TNF, IFNγ, and IL-6 and also atrogen expression levels, ⑮–⑰ chemokines, IL-10, TGFβ mediators, and viral genomic RNA are still detected, indicating an incomplete viral clearance. ⑱ Long-term immune activation results in a reduction in CSA by a UPS-independent mechanism. Symbols indicates: 
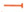
 block/inhibition; 
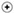
 positive regulation; 

 open ended questions.

**Table 1 ijms-25-08909-t001:** Primers sequences.

Gene	Fw (5′-3′)	Rv (5′-3′)
CHIKV	AAA GGG CAA ACT CAG CTT CAC	GCC TGG GCT CAT CGT TAT TC
CHIKV–FAM	/56-FAM/ CGC TGT GAT ACA GTG GTT TCG TGT G/ 3BHQ_1	
MAYV	CCT TCA CAC AGA TCA GAC	GCC TGG AAG TAC AAA GAA
MAYV–FAM	/56-FAM/ CAT AGA CAT CCT GAT AGA CTG CCA CC/ 3BHQ_1	
Atrogin-1	AGA AAA GCG GCA CCT TCG	CTT GGC TGC AAC ATC GTA GTT
MuRF1	GAG AAC CTG GAG AAG CAG CTC AT	CCG CGG TTG GTC CAG TAG
TNF-α	CCT CAC ACT CAG ATC ATC TTC TCA	TGC TTG TCT TTG AGA TCC ATG C
IFN-γ	AGC AAC AGC AAG GCG AAA A	CTG GAC CTG TGG GTT GTT GA
IL-6	TCA TAT CTT CAA CCA AGA GGTA	CAG TGA GGA ATG TCC ACA AAC
IL-1β	GTA ATG AAA GAC GGC ACA CC	ATT AGA AAC AGT CCA GCC CA
KC	CAC CTC AAG AAC ATC CAG AGC	AGG TGC CAT GAG AGC AGT CT
MCP-1	GTC CCC AGC TCA AGG AGT AT	CCT ACT TCT TCT CTG GGT TG
RANTES	GTG CCC ACG TCA AGG AGT AT	CCT ACT TCT TCT CTG GGT TG
TGF-β	GAC CGC AAC AAC GCC ATC TA	AGC CCT GTA TTC CGT CTC CTT
IL-10	TAA GGG TTA CTT GGG TTG CCA AG	CAA ATG CTC CTT GAT TTC TGG GC
Gapdh	AGG TCG GTC TGA ACG GAT TTG	TGT AGA CCA TGT AGT TGA GGT CA
β-actin	GAC GTT GAC ATC CGT AAA	GTA CTT GCG CTC AGG AGG AG

## Data Availability

Individual results were presented in dots at figures. Data will be made available for analysis upon request to iranaiamiranda@micro.ufrj.br.

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
