# Peer review of "Chikungunya and Mayaro Viruses Induce Chronic Skeletal Muscle Atrophy Triggered by Pro-Inflammatory and Oxidative Response"

_ijms, 2024, doi:10.3390/ijms25168909_

Round 1
Reviewer 1 Report
Comments and Suggestions for Authors
Manuscript by da Silva et al. analyses in-depth mechanisms of pathogenesis of two alphavirus infections, caused by Chikungunya and Mayaro viruses. Authors focus on the infection-associated skeletal muscle atrophy as a key part of the infectious process. Authors performed thorough morphological analysis of the progression in muscle disorders at early and late stages of infection. Also, they followed by qRT-PCR a virus genomic RNA expression and multiple host cytokine mRNA expression levels in muscle during the development of infectious process. Importantly, authors applied in their experiments two drug substances, namely, infliximab to bock TNF and monomethyl fumarate to activate Nrf2 protein as an antioxidant inducer. In both cases, they achieved a significant reduction in virus-associated muscle atrophy.
The paper is very logical, clear and perfectly written. Methods in use are highly relevant, results are convincing and comprehensible. Undoubtedly, this manuscript is of high novelty and soundness. Obviously, when published, it will have a strong scientific impact.
I noticed a few minor flaws in the manuscript.
Line 293: “…intense loss of muscle mass loss…” Apparently, the second “loss” here is redundant.
Lines 449-450: “Paraffin-embedded tissue sections of 5 mm were prepared…” I suppose, authors mean here a thickness of the section 5 µm, not 5 mm.
Line 451: “…optical microscopy with a magnification of 10 X…”. Clearly, the real magnification in images is much higher than 10 X, so, I think, authors mean here a microscope objective 10x. Please re-write this as “…optical microscopy with an objective 10 X…” Also, in principle, you could skip any mention of magnification in Methods, because scale bars in images give all the necessary information.
In general, after checking these small things in the manuscript, it can be published. It was a big pleasure for me to read and review it.
Reviewer 2 Report
Comments and Suggestions for Authors
The authors use a mouse model to study the acute and chronic impacts of MAYV and CHIKV replication on skeletal muscle (SkM) fiber structure, composition, and repair after injury. The authors documented destruction of muscle fiber and loss of muscle mass as a result of virus replication. Inflammatory responses resulted in not only acute muscle loss, but also chronic muscle atrophy. The authors go on to show that muscle mass waste and atrophy could be ameliorated by blocking TNF and inducing an antioxidant response. Overall, the results are convincing, and the manuscript is well organized and written. Specific suggestions for improvement are listed below.
1. The authors use a mouse model for their studies but no where do they justify the use of this model. A brief description of this model with appropriate references and a discussion of the pros and cons of this model in comparison to human disease progression is warranted.
2. The dark color scheme the authors use for the figures make it difficult to differentiate among the different viruses and mock. See for example Fig. 2D-E. Perhaps use a color scheme with a little more contrast to help the reader easily distinguish the different viruses.
3. Fig. 2B and 2F: all the bars are black instead of different colors as in Fig. 3.
4. Fig. 3A: the font is too small to be read easily.
5. Fig. 6F: inclusion of a mock treatment may help give context to the results.
6. Line 124: “comparison” should be “compare”.
7. Line 153: “accompanied” should be “imaged”.
8. Line 154: the phrase “combined to volume reconstitution” is difficult to understand. Please rephrase.
9. Lines 164-166: This sentence is hard to follow. Please rephrase.
10. Line 169: “unbalance” should be “imbalance”.
11. Line 171: This heading should read as follows: Skeletal muscle atrophy and genomic RNA persist at late phase of infection.
12. Line 187: “accompanied” should be “recorded”.
13. Line 301: “along life” perhaps should read as “throughout life”?
14. Line 311: “not complete” should read as "incomplete”.
15. Line 329: “SARS-CoV” should be “SARS-CoV-2”.
16. Line 344: “that would be determinant” is grammatically incorrect. Please fix.
17. Line 414: “105 or 106 pfu. Do you mean 105 or 106 pfu?
Comments on the Quality of English LanguageThe paper is very well written, however minor edits to language required.
Round 2
Reviewer 1 Report
Comments and Suggestions for Authors
The paper can be published in its present form.
Author Response
Thanks for the review
Best Regards